# Microdialysis on Ex Vivo Porcine Ear Skin Can Validly Study Dermal Penetration including the Fraction of Transfollicular Penetration—Demonstrated on Caffeine Nanocrystals

**DOI:** 10.3390/nano11092387

**Published:** 2021-09-14

**Authors:** Anna Lena Klein, Markus Lubda, David Specht, Sung-Min Pyo, Loris Busch, Jürgen Lademann, Martina C. Meinke, Ingeborg Beckers, Jörg von Hagen, Cornelia M. Keck, Alexa Patzelt

**Affiliations:** 1Center of Experimental and Applied Cutaneous Physiology, Department of Dermatology Venereology and Allergology, Freie Universität Berlin, Humboldt-Universität zu Berlin, and Berlin Institute of Health, Charité—Universitätsmedizin, 10117 Berlin, Germany; loris.busch@charite.de (L.B.); juergen.lademann@charite.de (J.L.); martina.meinke@charite.de (M.C.M.); alexa.patzelt@charite.de (A.P.); 2Merck KGaA, 64293 Darmstadt, Germany; markus.lubda@merckgroup.com (M.L.); joerg.von.hagen@merckgroup.com (J.v.H.); 3Department of Pharmaceutics and Biopharmaceutics, Philipps-Universität, 35037 Marburg, Germany; david.specht@pharmazie.uni-marburg.de (D.S.); ck@ckc-berlin.de (C.M.K.); 4BASF SE, R&D Pharma Ingredients, 67063 Ludwigshafen, Germany; sung-min.pyo@basf.com; 5Department II, Beuth University of Applied Sciences, 13353 Berlin, Germany; beckers@beuth-hochschule.de

**Keywords:** transfollicular penetration, microdialysis, IVIVC, caffeine, drug delivery system, nanotechnology

## Abstract

Common ex vivo methods for penetration investigations often fail to monitor transfollicular penetration appropriately. In the present investigation, the validity of dermal microdialysis on the ex vivo porcine ear skin to investigate penetration kinetics, including transfollicular penetration, was studied. In setup A, a caffeine nanocrystal formulation was compared to a non-particular caffeine gel formulation. In setup B, two caffeine nanocrystal formulations of different sizes (200 nm, 700 nm) were compared to each other. Microdialysis samples were collected for 46 h. After sampling, the skin layers were separated, homogenized, and caffeine was quantified in all samples. In setup A the area under the curve (AUC) after crystal gel formulation application was 12 times higher than after non-particular formulation application. Setup B showed an increased AUC of 42% in the microdialysis data when the 700 nm caffeine crystals were applied compared to the 200 nm crystals. The microdialysis data was supported by the separation, homogenization and extraction data. Microdialysis performed on ex vivo porcine ear skin is a novel experimental setup. It is of high interest for further investigations since it is able to also capture the impact of follicular and transfollicular penetration kinetics as no other ex vivo setup can.

## 1. Introduction

New topical formulations promising improved drug delivery are constantly being developed. Often, ex vivo testing of new formulations is mandatory before an ethically acceptable in vivo study can be performed. The aim of ex vivo studies is always to replicate the in vivo situation as closely as possible. Optimally, the results of such studies are well comparable with the expected in vivo results. Methods that are used in the kinetic study of topically applied substances are the Franz diffusion cell and dermal microdialysis. However, the Franz diffusion cell is the most commonly used method for such investigations and is suitable for many kinds of penetration investigations [1,2]. There are great efforts to gain a fundamental understanding of the skin barrier. This is especially true when it comes to new developments in the field of nanotechnological products [3]. Mathematical models have already been used to describe many aspects of the penetration of small lipophilic molecules. For macromolecules, however, the mathematical models postulate lower permeation than can be observed in experiments [4]. This could be due to follicular and transfollicular penetration, which could be more important for such molecules. The influence of the hair follicle at the penetration in the simulation could be shown by Kattou et al. [5]. When it comes to mimic follicular and transfollicular penetration ex vivo, dermal microdialysis becomes superior in comparison to Franz diffusion cell as discussed by Klein et al. [6]. This is particularly important for formulations that are designed to specifically utilize follicular and transfollicular penetration for increased drug delivery.

More precisely this is the case for formulations that use drug vehicles, such as nanocarriers, to increase the bioavailability of the active pharmaceutical ingredient (API). Nanocarriers above a size of 20–40 nm cannot pass an intact skin barrier, but deliver APIs to their target site and increase their bioavailability [7]. Thereby follicular penetration can be enhanced to create a depot of the drug within the hair follicle. It was shown that especially particulate nanosized structures penetrated well into the hair follicle, comparable to a mechanically driven transport process known as the ratchet effect [8]. The best results were achieved by using nanoparticles, with approximately 700 nm in diameter [9]. Recently it was shown that excipients can be used to control whether the API should preferably be delivered by passive diffusion or as a drug reservoir by depositing API in the hair follicles [10]. Inside the hair follicle, the drug can be released by, e.g., diffusion or several trigger mechanisms [11] so that the API can pass the intrafollicular barrier without its vehicle [12]. This represents an effective penetration pathway as the barrier function inside the hair follicle is physiologically reduced [10].

The in vivo efficiency of topical formulations that target the hair follicle and penetrate transdermally is easy to quantify by measuring plasma concentrations [13]. It is a greater challenge to determine drug concentrations in dermal tissue by routine methods in vivo as well as ex vivo. To investigate the penetration kinetics of a topically applied formulation in vivo dermal tissue dermal microdialysis is the method of choice [14]. Dermal microdialysis represents a method for extracting endogenous and exogenous substances from different tissues, which was first used in 1966 for neurological tissue examinations [15,16]. With the help of dermal microdialysis, the penetration behavior of numerous topically applied substances has already been investigated [17,18,19].

Since not all studies can be performed in vivo, a reliable ex vivo sampling method to investigate transfollicular penetration is required. Additionally, for ex vivo penetration studies, dermal microdialysis was used and is a promising tool to also determine the kinetical impact of the transfollicular penetration. Döge et al. [20] performed a penetration study on excised human skin by microdialysis. Here the penetration of dexamethasone nanocrystals was investigated and compared to a particle-free dexamethasone cream. An increased concentration of dexamethasone was collected at sites where the nanocrystal formulation had been applied. However, excised human skin is not a well-fitting skin model to investigate transfollicular penetration since in excised human skin, the elastic fibers around the hair follicle contract and therefore reduce the follicular reservoir [21].

It was shown that porcine ear skin is a well suitable skin model for penetration studies [22,23]. This becomes superior when it comes to the investigation of follicular and transfollicular penetration since the skin remains fixed on the cartilage. Klein et al. [6] recently showed that microdialysis performed on porcine ear skin is capable to monitor the transfollicular penetration kinetics of caffeine. To investigate the effect of transfollicular penetration, caffeine can be used as an API since previous in vivo and ex vivo studies showed that the follicular penetration pathway influences the penetration kinetics of caffeine [24,25]. Furthermore, the OECD guidelines recommend caffeine for skin penetration studies [26].

In the context of the present investigation, the validity of dermal microdialysis on the ex vivo porcine ear skin to investigate transfollicular penetration was studied in greater detail. The study at hand was focused on the formulation-based kinetic effects on the penetration of caffeine and the correlation between particle size and bioavailability. Doing this, identical formulations as used in an in vivo study by Breuckmann et al. [13] were investigated as well as a comparable caffeine gel formulation. The separation of the skin layers, their homogenization and the extraction of the API from the tissue were performed for all areas that were investigated by dermal microdialysis and performed after 46 h. It was investigated to what extent an ex vivo microdialysis setup is able to mimic the in vivo situation. A large difference between the formulations in setup A was expected, while a comparably small difference was expected in setup B.

## 2. Materials and Methods

### 2.1. Materials

#### 2.1.1. Formulations: Caffeine Nanocrystals and Saturated Caffeine Gel

The different caffeine formulations (caffeine gel, caffeine nanocrystals of small and medium particle size) were produced according to previously developed protocols [27]. The gel contained 2.5% (*w*/*w*) caffeine (Merck KGaA, Darmstadt, Germany) dissolved in a mixture of 30% ethanol (Fisher Scientific, Loughborough, UK) and 70% propylene glycol (Caesar & Lorentz GmbH, Hilden, Germany) to which 2% Carbopol^®^ 981 (Lubrizol Advanced Materials Europe BVBA, Brussels, Belgium) was added while mixing with mortar and pestle to obtain a highly viscous gel [13].

The caffeine nanocrystals contained 20.0% (*w*/*w*) caffeine, and the dispersion medium was similar to that of the caffeine gel (mixture of 30% ethanol, 70% propylene glycol with 2% Carbopol^®^ 981 as gelling agent). The nanocrystals were produced by adding coarse caffeine crystals to the gel phase. The mixture was high speed stirred (Ultra-Turrax T25 (Janke & Kunkel GmbH, Cologne, Germany) at 5000 rpm for 30 s, and the obtained pre-suspension was subjected to bead milling at 2000 rpm (PML 2, Bühler AG, Appenzell, Switzerland). Milling was performed in discontinuous mode in a small volume milling chamber with an active volume process zone of 222 mL. The bead size was (0.4–0.6) mm (Yttria stabilized zirconium oxide beads, Hosokawa Alpine, Augsburg, Germany), the suspension to bead ratio was 50:50 (*v*/*v*). The milling time was set to 15 min for the medium-sized caffeine nanocrystals and to 3 h for the small-sized nanocrystals [13,28]. The particle size and size distribution of the nanocrystals were determined by dynamic light scattering (DLS; Zetasizer Nano ZS, Malvern Panalytical Ltd., Malvern, UK). DLS analyses the diffusion constant D of the particles, which was used to calculate the hydrodynamic diameter (z-average) of the particles. In addition, the particle size distribution was estimated by calculation of the polydispersity index (PDI) and by converting the DLS size results into the median volume-weighted diameters d(v)0.1–d(v)0.9. The composition of the formulations and their physico–chemical properties are summarized in Table 1.

#### 2.1.2. Skin Model: Porcine Ear Skin

All experiments were performed using ex vivo porcine ear skin (6-months old German domestic pigs) provided by a local slaughterhouse. The experiments were authorized by the Commission of Consumer Protection and Agriculture, District Dahme-Spreewald, Germany. After arrival, the porcine ears were cleaned with cold water, dried using paper towels, and stored at 8 °C in a hydrated chamber overnight. The next day, hairs were cut short, and the experiments were performed. The used porcine ears had an average follicular density of 12 follicles per cm^2^ in the investigated areas in both setups.

### 2.2. Methods

#### 2.2.1. Experimental Setups

The study was separated into two setups. In setup A, a saturated caffeine gel formulation was compared to medium-sized caffeine nanocrystals which was expected to be superior to the gel formulation. Setup B monitored the influence of the nanocrystal size on the penetration by comparing two sizes of caffeine nanocrystals. In this setup, small nanocrystals of about 180 nm in size and medium particles of about 700 nm in size were used (c.f. Table 1). In setup B, the dispersion medium (gel phase) was used as a negative control formulation. The two setups of the study are illustrated in Figure 1. The overall penetration and the pharmacokinetics of the investigated formulations are monitored by separation, homogenization and extraction of the API after 46 h and dermal microdialysis for 46 h in both setups. Figure 2 shows the sampling methods.

#### 2.2.2. Skin Preparation and Application of Formulations

For each investigated site a 1 × 1 cm^2^ area was prepared. A silicone barrier (WINDOW COLOR, Marabu GmBH & Co. KG, Bietigheim-Bissingen, Germany) was applied around the skin areas to prevent lateral spreading, according to Jacobi et al. [29]. A total of 10 µL/cm^2^ of area-related formulation was applied to the skin area by a displacement pipette. A massage was executed by a glove-covered cotton swab for one minute using a massage frequency of approximately 5 Hz, as suggested by Busch et al. [30]. The tip of the pipette and the glove-covered cotton swab were collected in 1 mL phosphate-buffered saline (PBS; pH 7.4, sterile, VWR LIFE SCIENCE, Lutherville, UK) for caffeine extraction. The experiment was performed on five ears from various donors.

#### 2.2.3. Dermal Microdialysis

Dermal microdialysis was used to analyze the caffeine concentration in dermal tissue. For this technique, a membrane is placed in the upper part of the dermis, constantly perfused and the perfusion medium is collected. The same material and general microdialysis setup as described by Klein et al. [6] were used. In setup A, the mean depth of the membrane was 0.32 mm with an average standard deviation of 0.05 mm while in setup B, this depth was 0.30 mm and the mean standard deviation was 0.06 mm.

Samples were collected over a period of 46 h with a total of 17 sampling times. At the beginning, the collection intervals were 10 min while at later times, they were larger and collection times up to 16 h apart. The exact intervals can be taken from Table 2. The area under the curve (AUC) was calculated based on the total caffeine collected in the dialysate within the collection time.

For the investigation of the recovered caffeine in the different skin layers, the overall collected caffeine by microdialysis was added to the collected caffeine in dermis since the membrane was placed in the upper part of the dermis.

#### 2.2.4. Skin Separation, Homogenization and Extraction after 46 h

After 46 h, the skin layers were separated to analyze how much caffeine penetrated. The separation of the skin layers (tape stripping, heat separation [31]), the homogenization and extraction were performed similarly as in the study by Klein et al. [6]. The only difference between the protocols used was that in the present study, more tape strips had to be taken to remove the remaining formulation from the skin surface. All tapes were included in the analysis. Thus, a total of 16 tapes were taken and extracted in 2 mL PBS. Furthermore, the samples were frozen until they were subjected to HPLC.

In order to determine the percentage of applied caffeine that can be recovered from the tissue, it was first measured how much caffeine of the different formulations is applied with the selected application method. For this purpose, each formulation was pipetted three times (n = 3) into corresponding reaction vessel containing 1 mL PBS with the displacement pipette and dissolved including the formulation residues at the pipette tip. The resulting solutions were analyzed by HPLC.

#### 2.2.5. Determination of Caffeine by HPLC

The HPLC was conducted using a VWR-Hitachi ELITE LaChrom system (VWR, Darmstadt, Germany) and caffeine was quantitatively determined with a DAD I-2450 detection unit (VWR, Darmstadt, Germany) at a wavelength of 272 nm and 30 °C. A Chromolith^®^ Performance RP-18e 100–4.6 mm (Merck KGaA, Darmstadt, Germany) column was used as stationary phase and an isochratic gradient with 90% water (Merck KGaA, Darmstadt, Germany) and 10% acetonitrile (Merck KGaA, Darmstadt, Germany) as mobile phase at a flow rate of 2.0 mL/min. The samples were mixed in an autosampler screw vial (VWR, Darmstadt, Germany) and analyzed with an injection volume of 10 or 30 µL. An internal caffeine standard solution confirmed the linear quantification of caffeine and a calibration standard (0.5–250.0 μg/mL) in PBS buffer was determined with a linear regression conformation of R2 > 0.99 and an accepted variation of less than 2% accuracy.

#### 2.2.6. Statistical Analysis

The calculation of mean values and standard deviations of the penetration and corresponding statistical tests (t-test or Wilcoxon test depending on the normal distribution of the dataset), were carried out via SPSS^®^ (Version 27, IBM Cooperation, New York, NY, USA). For *p*-values less than 0.05 the null hypothesis was rejected.

## 3. Results

### 3.1. Setup A—Comparison of Caffeine Gel and Caffeine Nanocrystal Gel Formulation

#### 3.1.1. Investigation of Caffeine Distribution in the Skin Layers by Separation, Homogenization and Extraction after 46 h

On average, 84% of the total applied caffeine was recovered from all extracted samples from the skin areas with gel application. In areas where the nanocrystals were applied, 77% of the total applied formulation was recovered (Figure 3a). A 25 times higher amount of caffeine was applied on skin areas that were treated with the nanocrystal gel formulation.

The caffeine obtained from the dermis, shown in Figure 3b, includes both the caffeine extracted from the dermal tissue by homogenization and the caffeine collected by microdialysis. While (21 ± 4) µg caffeine was extracted from the dermal tissue in areas where the gel formulation was applied, (202 ± 95) µg caffeine was extracted from the dermal tissue where the nanocrystals were applied (*p* < 0.05).

#### 3.1.2. Investigation of Pharmacokinetics by Dermal Microdialysis for 46 h

After 10 min, caffeine was detectable in the dialysate from both formulations. However, the maximum mean caffeine concentration obtained from skin areas treated with the nanocrystal gel formulation was 11 times higher (46.5 µg/mL) than the maximum mean caffeine concentration of the gel-treated skin areas (3.8 µg/mL). Within the first hour, the mean caffeine concentration in the gel-treated skin areas increased to 3.1 µg/mL, while skin areas treated with the nanocrystal gel formulation showed a mean caffeine concentration of 40.1 µg/mL. After two hours, the differences between the two caffeine application forms always became significant (Figure 4a). Furthermore, the median AUC was 12 times larger for the nanocrystal gel formulation than for the gel formulation (Figure 4b), *p* < 0.05).

### 3.2. Setup B—Comparison of Two Sizes of Caffeine Nanocrystals

#### 3.2.1. Investigation of Caffeine Distribution in the Skin Layers by Separation, Homogenization and Extraction after 46 h

The total recovery rate of applied caffeine was 89% from skin areas where the 200 nm nanocrystals were applied, and 72% from the skin areas treated with 700 nm nanocrystals (*p* < 0.05) (Figure 5a).

The amount of caffeine shown in Figure 5b includes caffeine extracted from the homogenized dermis and from the dermis by microdialysis. While (10 ± 6) µg caffeine was extracted from the dermal tissue in areas where the 200 nm nanocrystal gel formulation was applied, (19 ± 14) µg caffeine was extracted from the dermal tissue where the 700 nm nanocrystals were applied (*p* < 0.05).

#### 3.2.2. Investigation of Pharmacokinetics by Dermal Microdialysis for 46 h

Both nanocrystal gel formulations showed a constantly slow rise of the caffeine concentration in the dialysate. At all times, the median concentration of caffeine was higher at skin areas where the 700 nm nanocrystals were applied. The maximal mean caffeine concentration was reached after collecting for 30 h (1800 min) and was 1.4 µg/mL for the 700 nm nanocrystals and 0.8 µg/mL for the 200 nm nanocrystals. After a collection time of 22 h (1320 min), the difference between the two nanocrystals is significant at all but one time points (Figure 6a). The median AUC of the 700 nm nanocrystals was 42% higher than the AUC of the 200 nm nanocrystals (Figure 6b, *p* < 0.05).

## 4. Discussion

The present study investigated to what extent ex vivo dermal microdialysis on porcine ear skin is able to mimic an in vivo investigation. The penetration kinetics of topically applied caffeine formulations was investigated by microdialysis over 46 h. The separation of the skin layers, their homogenization and the extraction of the API from the tissue was also performed for all areas and performed after 46 h. In setup A, the caffeine penetration from a saturated caffeine gel formulation was compared to a caffeine nanocrystal gel formulation to investigate the general effect of using nanocrystals compared to a particle-free formulation when having the follicular pathway available. A large difference was expected in setup A when comparing a particle-free to a particle-containing formulation. The influence of the particle size on bioavailability was investigated in setup B. Two caffeine formulations containing nanocrystals of different sizes (200 nm and 700 nm) were compared. The induced effect was expected to be comparably small. Since the same formulations were used in an in vivo study, it is to determine how well an ex vivo microdialysis setup can mimic the in vivo situation.

The separation, homogenization and extraction data, as well as the microdialysis data from setup A, clearly demonstrate the advantage of using a particle-based formulation. More caffeine was collected in the microdialysis dialysate at every time point in areas where the nanocrystal gel formulation was applied. The AUC and the total extracted caffeine from the dermis were significantly higher at sites where the nanocrystal gel formulation was applied. It needs to be remembered that on the compared areas in setup A the amount of caffeine applied per area was not always the same. However, the applied gel formulation was saturated and therefore it would have been impossible to apply a greater amount of caffeine in the used gel base. The nanocrystals are therefore necessary to reach the desired increase in bioavailability

Increased bioavailability of an applied API related to particle-based formulations was demonstrated before. For example, Döge et al. [20] compared a dexamethasone nanocrystal gel formulation (0.05% (*w*/*v*)) to a dexamethasone cream formulation (0.05% (*w*/*v*)) using dermal microdialysis on excised human skin. The impact of using nanocrystals was not as high as in the here presented study. The nanocrystals increased the concentration of dexamethasone approximately 5-fold in the dialysate after 12 h. In the present study, an approximately 15-fold increase could be observed when comparing the nanocrystal gel formulation to the gel formulation at this time point. On the one hand, this could be due to the particle size since the particles Döge et al. used were 200 nm on average. Furthermore, another API was monitored, and other formulations (nanocrystal gel formulation vs. control formulation) were used in their study. On the other hand, the study was performed on excised human skin. Patzelt et al. [21] demonstrated that in excised human skin the elastic fibers around the hair follicle are contracted and therefore reduce the follicular reservoir by up to 90%. Overall, it can be stated that both, the here presented study and the study by Döge et al. [20], showed the advantage of using particle-based formulations. However, the data obtained on porcine ear skin showed the advantage of particles to a larger extent, probably since the particles could better penetrate the hair follicle on porcine ear skin than on excised human skin. Additionally, the other reported factors such as the different formulations (nanocrystal gel formulation vs. control formulation) and the different APIs could have supported the observed difference.

Abd et al. [32] compared two caffeine nanoemulsions to an aqueous caffeine control solution. The experiments were performed on Franz diffusion cells. In the Abd et al. study, a 36-fold and 42-fold increased caffeine permeation was shown over 24 h for the two nanoemulsions compared to the aqueous caffeine solution. The monitored increase in concentration was higher than the increase observed in the here presented study where the median AUC was increased 10 times, and the total caffeine in the dermis was on average 8 times higher than by using the gel formulation. A likely explanation for these differences could be the different methods used. While microdialysis was performed on intact porcine ear skin that was still attached to the cartilage, the Franz diffusion cell study required excision of the skin and removal of the fatty tissue underneath the dermis. Since hair follicles can reach deeply into the dermal tissue, the follicular barrier might be disrupted when preparing the human skin for Franz diffusion cell experiments [33,34]. If the follicular barrier is disturbed and a particulate substance is applied that uses the follicular pathway more than a non-particulate substance, it is comprehensible that the influence of a disturbed follicular barrier is more pronounced for the particulate formulation. Therefore, in the Franz diffusion cell study, the particles probably faced an even weaker barrier after penetrating into the hair follicle than in the present microdialysis study.

Even though various studies have already shown an influence of nanoparticles on penetration, this is the first ex vivo setup that is truly able to investigate transfollicular penetration without follicular contraction and potential barrier disruption by skin preparation.

When analyzing the AUC of the microdialysis data in setup B, it can be seen that more caffeine became bioavailable when the larger 700 nm caffeine nanocrystals were applied. In the in vivo study by Breuckmann et al. [13], the AUC was determined as well. There, an increase of 82% was achieved over 120 h when using the 700 nm nanocrystals instead of the 200 nm nanocrystals. In the present study, an increase of 72% was achieved over 48 h for the same scenario on ex vivo porcine ear skin. These results show that the used ex vivo setup is well comparable to the already performed in vivo study, especially when considering the different time frames and variation of methods.

The microdialysis data obtained from setup B showed that both nanocrystal sizes delivered caffeine in a similar kinetic curve. However, in the median, the absolutely collected caffeine was higher for the larger nanocrystals at every time point. The median AUC also showed that the larger nanoparticles were able to provide a higher bioavailability of the API. This result was expected in trend and in height since nanoparticles of the size around 700 nm were previously demonstrated to penetrate deeper into the hair follicles than smaller nanoparticles [9].

Furthermore, the total uptake of caffeine in the dermis (extracted from the homogenized dermis and microdialysis dialysates) is obviously a good argument for using the larger nanoparticles to enhance the transport of caffeine to dermal tissue. In addition to the observation that more caffeine could be found in the dermis after the application of the large nanocrystals, another interesting information could be obtained from the homogenization data. It was analyzed how much of the total applied caffeine could be recovered. Since the same procedure was used for the separation, homogenization, and extraction of caffeine from the tissue in the areas of both nanocrystal sizes, it should be assumed that the same amount of caffeine can be extracted. Contrary to this assumption, significantly less caffeine was recovered in areas where the large nanocrystals were applied. This difference in recovery could be attributed to the fact that the setup used is an open system. It is known that caffeine can penetrate transdermally [35]. Therefore, it can be assumed that the lower recovery of caffeine in sites where the large nanocrystals were applied is correlated to an overall higher penetration and therefore also a higher transdermal penetration. This supports the advantage of using large nanocrystals.

## 5. Conclusions

In the framework of the present study, we were able to show a higher dermal bioavailability of caffeine when applied in nanocrystalline form. Furthermore, a higher dermal bioavailability was obtained with 700 nm nanocrystals in comparison to 200 nm nanocrystals. Overall, it can be stated that the expected correlation in bioavailability between the nanocrystal’s sizes on the one hand, and between nanocrystals and caffeine gel formulation on the other hand, could be monitored by microdialysis itself. The respective data could be supported by the separation, homogenization, and extraction method. Therefore, it can be concluded that microdialysis performed on porcine ear skin is a valid ex vivo tool to investigate dermal and transfollicular penetration of topically applied formulations. Results obtained by the presented method are well comparable to in vivo data as well as to other ex vivo methods that are less suitable for transfollicular investigation than the method presented here. The method’s outstanding and novel character is that it initially can capture the impact of follicular and transfollicular penetration on the overall penetration in an ex vivo setup. By keeping the skin on the cartilage, no shrinking of the follicles is induced, and the bottoms of the hair follicles are not cut off as can happen when performing penetration studies using Franz diffusion cells.

## Figures and Tables

**Figure 1 nanomaterials-11-02387-f001:**
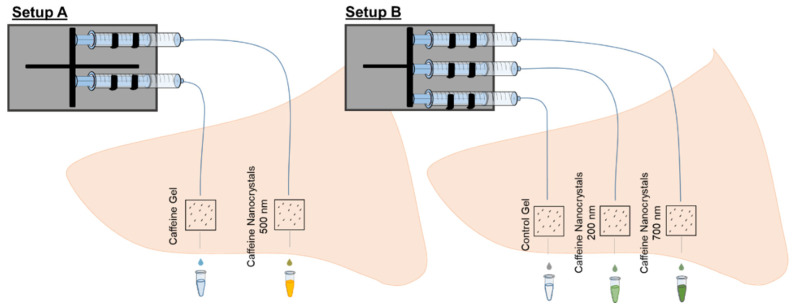
Setup A: Two skin areas were prepared on each porcine ear. A gel formulation and nanocrystal gel formulation were applied. Setup B: Three skin areas were prepared on each porcine ear. A control formulation and formulations containing 200 nm nanocrystals and 700 nm nanocrystals were applied.

**Figure 2 nanomaterials-11-02387-f002:**
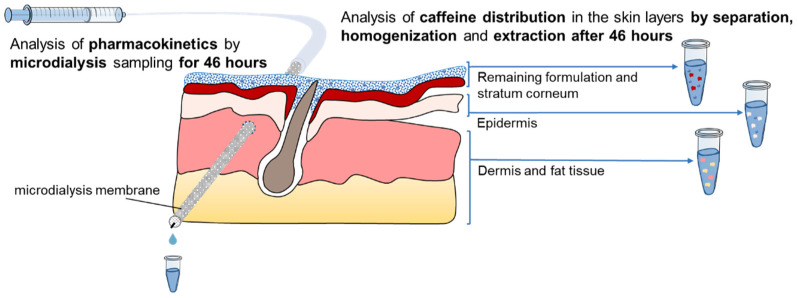
Dermal microdialysis and skin separation, homogenization and extraction were performed in parallel on the porcine ear skin. The dermal microdialysis is able to capture the pharmacokinetics of the applied formulations, while the skin separation, homogenization and extraction technique captures the caffeine distribution in the skin layers at one time point.

**Figure 3 nanomaterials-11-02387-f003:**
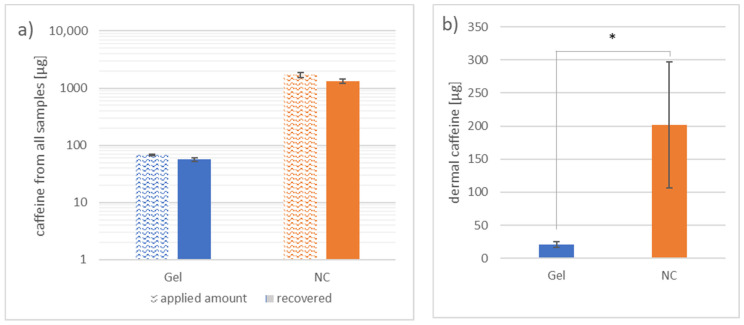
Graph (**a**) shows the overall collected and extracted caffeine from areas where caffeine gel or caffeine nanocrystal gel formulation (NC) was applied in correlation to the total amount of caffeine applied on a logarithmic scale. Graph (**b**) presents the total amount extracted from the dermis by homogenization and microdialysis. Differences between the two areas of *p* < 0.05 were considered significant (*). n = 5 (applied amount: n = 3).

**Figure 4 nanomaterials-11-02387-f004:**
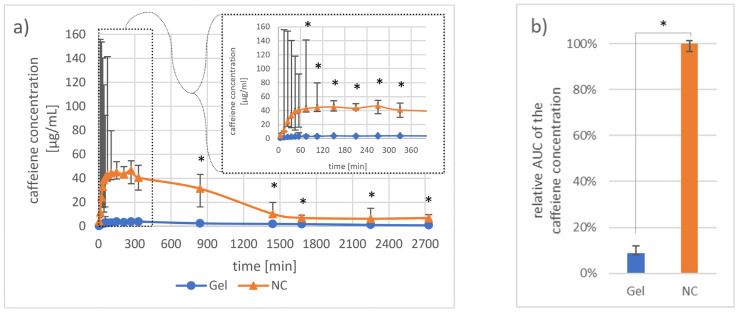
Graph (**a**) presents the median ± one quartile caffeine concentration sampled by microdialysis from the two investigated areas on porcine ear skin compared at different time points. Graph (**b**) shows the median ± one quartile area under the curve from the obtained caffeine concentration data. The gel- and the nanocrystal gel formulation (NC) were compared in both graphs. Differences between the two areas of *p* < 0.05 were considered significant (*). n = 5.

**Figure 5 nanomaterials-11-02387-f005:**
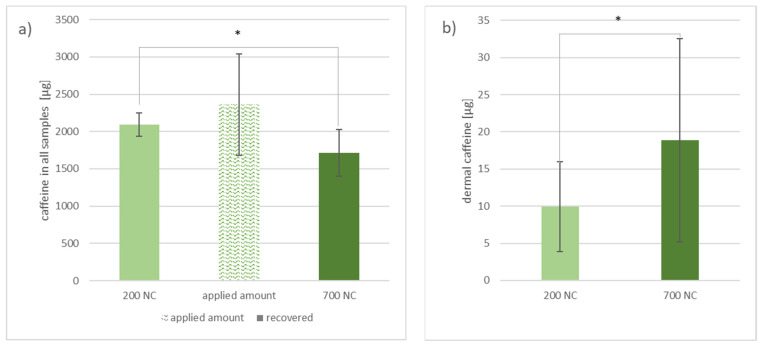
Graph (**a**) shows the overall collected and extracted caffeine from areas where either 200 nm nanocrystals (200 NC) or 700 nm nanocrystals (700 NC) were applied on porcine ear skin in correlation to the total amount of caffeine applied. Graph (**b**) presents the total amount extracted from the dermis by homogenization and microdialysis. Differences between the two areas of *p* < 0.05 were considered significant (*). n = 5 (applied amount: n = 3).

**Figure 6 nanomaterials-11-02387-f006:**
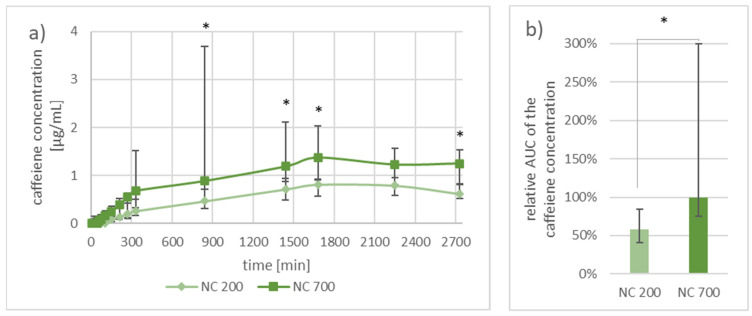
Graph (**a**) presents the median ± one quartile caffeine concentration sampled by microdialysis from the two investigated areas on porcine ear skin with 200 nm nanocrystals (200 NC) and 700 nm (700 NC) at different time points. Graph (**b**) shows the median ± one quartile area under the curve from the obtained caffeine concentration data. Two nanocrystal gel formulations were compared to each other in both graphics. Differences between two areas of *p* < 0.05 were considered as significant (*). n = 5.

**Table 1 nanomaterials-11-02387-t001:** Overview of the formulations used in the study.

Formulations	Composition	z-Average [nm]	PDI	d(v)0.1 [nm]
				d(v)0.5 [nm]
				d(v)0.9 [nm]
caffeine gel	caffeine 2.5% (*w*/*w*) gel phase * add 100.0% (*w*/*w*)	n.a. (formulation contained dissolved caffeine)		
caffeine nanocrystals (setup A)	caffeine 20.0% (*w*/*w*) gel phase * add 100.0% (*w*/*w*)	535	0.66	351
				462
				609
caffeine nanocrystals small size (setup B)	caffeine 20.0% (*w*/*w*) gel phase * add 100.0% (*w*/*w*)	178	0.59	176
				225
				288
caffeine nanocrystals medium size (setup B)	caffeine 20.0% (*w*/*w*) gel phase * add 100.0% (*w*/*w*)	689	0.74	405
				531
				691

* Gel phase consisted of 68.6% (*w*/*w*) propylene glycol, 29.4% (*w*/*w*) ethanol and 2.0% polyacrylate as gelling agent.

**Table 2 nanomaterials-11-02387-t002:** Overview of microdialysis collection times.

Sample	Collection Time	Duration of Collecting
1	10 min	10 min
2	20 min	10 min
3	30 min	10 min
4	40 min	10 min
5	50 min	10 min
6	1 h	10 min
7	1.5 h	30 min
8	2 h	30 min
9	3 h	1 h
10	4 h	1 h
11	5 h	1 h
12	6 h	1 h
13	22 h	16 h
14	26 h	4 h
15	30 h	4 h
16	45 h	15 h
17	46 h	1 h

## Data Availability

The data presented in this study are available in Table 1 as well as in Figure 3, Figure 4, Figure 5 and Figure 6. Further data that support the findings of this study are available from the corresponding author upon reasonable request via email.

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
