# Peer review of "Microdialysis on Ex Vivo Porcine Ear Skin Can Validly Study Dermal Penetration including the Fraction of Transfollicular Penetration—Demonstrated on Caffeine Nanocrystals"

_nanomaterials, 2021, doi:10.3390/nano11092387_

Round 1
Reviewer 1 Report
Nanomaterials-1340291
Microdialysis on ex vivo porcine ear skin can validly study dermal penetration including the fraction of transfollicular penetration – demonstrated on caffeine nanocrystals
Comments and Questions:
The advantage of using particle-based formulations were demonstrated. The data obtained on porcine ear skin showed the advantage of particles to a larger extent, probably since the particles could better penetrate the hair follicle on porcine ear skin than on excised human skin.
- Nanocrystals with optimized particle size (650-750 microns) can preferentially accumulate in hair follicles increasing the penetration into surrounding cells may require more references with direct evidences, such as biological specimens or assays, to support.
- Please cite and include the papers: Baroli, B., “Penetration of Nanoparticles and Nanomaterials in the Skin: Fiction or Reality?” J Pharm Sci, 99(1) 21-50 (2010) and Mitragotri, S., “Engineering Approaches to Transdermal Drug Delivery: A Tribute to Contributions of Prof. Robert Langer,” Skin Pharmacol Physiol 2013;26:263–276 in the INTRODUCTION section. How can these two papers be related to the use of nanocrystals?
- Apposed to high pressure homogenization (Int J Pharm 470 (2014) 141-150), pearl milling operated at 5000 rpm and bead milling 2000 rpm could be hard to scale up. Please comment.
- The crystallinity of “nanocrystals” should always be checked by PXRD and not just optical microscopy. Most likely, the crystallinity is significantly reduced by powerful mechanical shear. Amorphous phase with higher solubility (or degree of crystallinity not just size) may also play a key role in the increase of caffeine “nanoparticle” penetration. Please clarify the effect of: size vs. crystallinity.
- Do the elastic fibers around the hair follicle contract (Is this inevitable?) only happen to all excised human skin but not normal human skin? If not, despite of the advantage of porcine ear skin over human skin in the experiments, should excised human skin be used for dermal penetration study because of its proximity to real-life applications?
Author Response
"Please see the attachment."

Reviewer 2 Report
The authors present an interesting study that proposes use of the porcine ear model for ex vivo microdialysis to study dermal drug penetration. This is the first time this strategy has been tested; the work is therefore of high interest for the scientific community.
The manuscript is well written, both overall aims and structure are clear, the methodological approach is sound and sufficient background information is presented for the reader.
Questions/missing information:
- Section 2.2.4.: please include a brief description of the exact tape stripping approach in the text (skin cleaning? first tapes rejected, or also analysed and included in analysis?)
- Legends Fig 3 and 5: please explain the number of experiments n stated here. In section 2.2.2. it is stated that n=5 (5 ears). What does "total n=3" and "applied amount n=3" refer to?
Author Response
"Please see the attachment."

Round 2
Reviewer 1 Report
Accept as it is.